# Education for Sustainability, Peace, and Global Citizenship: An Integrative Approach

Constantinos Yanniris 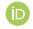

Department of Primary Education, University of Ioannina, T.K. 45110 Ioannina, Greece; constantinos.yanniris@mail.mcgill.ca

**Abstract:** The complex nature of contemporary challenges requires a culture of cooperation between academic disciplines. However, to what extent do educational systems prepare students to think beyond the boundaries of austerely defined and often entrenched academic fields? UNESCO has successively called for Environmental Education, Education for Sustainable Development, and Education for Global Citizenship to incorporate complex socio-environmental issues into mainstream education. Despite the presence of strong institutional support by governments and international organizations, the introduction of these interdisciplinary approaches into actual educational settings has been slow. With no intention to underestimate the pertinence and agency of strong political will in promoting educational change, we explore the presence of deeper, epistemological issues that may account for the generally slow progress of interdisciplinary pedagogies. To elaborate on this discussion, we focus on pragmatic solutions that can promote the integration of environmental, sustainability, and global citizenship education into the existing educational ethoi.

**Keywords:** Environmental Education; Education for Sustainable Development; Global Citizenship Education; learning outcomes; knowledge integration; connectedness to nature; epistemology; systems of knowledge

## 1. Introduction

Today, a growing global population demands the allocation of ever more natural resources to meet a seemingly insatiable desire for consumption. This demand strains the availability of natural resources and increases competition over their use. As a result, humanity may face a future of irreversible environmental damage and continuing conflict over access to dwindling environmental resources [1]. Education can help alleviate this disaster by demonstrating (a) the effects of our civic and economic choices on the state of the natural environment and (b) the impact of environmental issues on the health and resilience of human societies. International organizations have successively called for Environmental Education, Education for Sustainable Development, and Education for Global Citizenship to incorporate these complex socio-environmental issues in mainstream education. These often-overlapping educational approaches were introduced in the 1970s, 1990s, and 2010s, respectively. The aim of this review essay is to assess the hitherto impact and future prospects of Environmental Education, Education for Sustainable Development, and Education for Global Citizenship. To this end, this essay firstly touches on the institutional history of the three educational approaches. Then, it presents real-world examples and discusses the pragmatic challenges that these approaches encountered upon their adaptation to actual educational settings. Lastly, the essay discusses whether the above-mentioned educational approaches have the inherent potential to induce the radical educational change that they proclaim—thus generating a deeper, transformational impact on teaching and learning practices.

In 1977, the world's first intergovernmental conference on Environmental Education was organized under the auspices of UNESCO and UNEP in Tbilisi. The declaration

adopted in the context of this first international conference on Environmental Education describes the field as "interdisciplinary and holistic in nature and application . . . an approach to education rather than a subject" [2]. The Tbilisi declaration expounds that "Environmental Education must adopt a holistic perspective which examines the ecological, social, cultural and other aspects of particular problems. It is therefore inherently interdisciplinary" [3]. Following the introduction of the term "sustainability" into the public discourse by the Brundtland Commission Report (1987), Education for Sustainable Development was presented at the United Nations Environmental Conference in Rio (1992) [4,5]. Sustainable development was defined as "development which meets the needs of the present without compromising the ability of future generations to meet their needs" [4]. Drawing on this definition, the political demand for Sustainable Development is viewed as a synthesis between environmental theory (the interest in conserving natural resources) and justice theory (a concern for the needs of present and future generations) [6]. Hence, Education for Sustainable Development placed renewed emphasis on the economic, social, and political aspects of the environmental problem. In the following decade, efforts to integrate environmental and sustainability themes into diverse educational curricula continued with initiatives such as the 2005–2014 UN Decade of Education for Sustainable Development [7].

Furthermore, the UN Sustainable Development Goals constitute a collection of 17 global goals, announced in December 2014 and adopted by the United Nations General Assembly in September 2015 [8]. With respect to education, Sustainable Development Goal 4 aims to "ensure inclusive and equitable quality education and promote lifelong learning opportunities for all" by the year 2030 [9].

More recently, UNESCO (2015) introduced Global Citizenship Education, a form of civic learning that involves students' active participation in projects that address global issues of a social, political, economic, or environmental nature [10]. In 2016, UNESCO mandated that "(i) global citizenship education and (ii) education for sustainable development, including gender equality and human rights, are mainstreamed at all levels in: (a) national education policies, (b) curricula, (c) teacher education and (d) student assessment" [11].

Despite the presence of consistent institutional support by international organizations, the introduction of these interdisciplinary pedagogies into actual educational settings proved challenging. Indicatively, four decades after its celebrated Tbilisi inauguration, the integration of Environmental Education into educational curricula continues to meet significant resistance [12]. Even in countries (and states) where a favorable political climate has led to the implementation of policies in support of educational initiatives linked to environmental concerns, the integration of environmental and sustainability education into the schooling system has not proceeded with the pace that many of us would have hoped for [13–15].

The introduction of Education for Sustainable Development was also accompanied by criticism that teachers and practitioners were left without guidance in a discourse that was becoming increasingly abstract and decontextualized from pedagogy and contexts of practice [16–18]. Furthermore, the UN Sustainable Development Goals were deemed as "unachievable", and the organization was criticized for "cockpit-ism", i.e., the "illusion that top-down steering by governments and intergovernmental organizations alone can address global problems" [19]. In response, UNESCO acknowledged this criticism and proceeded to specify the implementation and monitoring policies on UN Goal 4 pertaining to education [11].

Moreover, contemporary approaches to Environmental Education, Education for Sustainable Development, as well as Global Citizenship Education have been criticized for reproducing colonial systems of power where a "we" in the Global North can learn about and solve the problems of a "them" in the Global South [20]. This type of critique deplores the superficial transfer of "successful" educational practices from the Global North to the Global South [21].

Contrasting the mentality of cockpit-ism and unilaterally imposed solutions, this review essay discusses three cases from outside the first tier of economic development, where environmental, sustainability, and global citizenship education programs have responded to the socioeconomic and educational needs of the local communities. These cases represent examples of interdisciplinary cooperation and synthesis in teaching and learning.

## 2. Examples from the Global South

This section discusses three cases that have produced encouraging results in Environmental, Sustainability, and Global Citizenship Education. All three examples that were selected to be presented in this article derive from the Global South. It was a purposeful decision to present cases of exemplary educational programs from the Global South in this review essay. There is indeed a growing interest in the literature concerning pedagogies developed in the Global South – consider, for example, the influential work of de Sousa Santos, Epistemologies of the South and the future [22]. Besides, there does not seem to be any pragmatic basis for the emphasis given by the relevant literature in featuring programs from the Global North [23]. Instead, research on program evaluation from countries of the Global North has returned unimpressive learning outcomes in Environmental and Sustainability Education [12].

While it too early for a coherent assessment of Global Citizenship Education programs (since this is the most recent of the three approaches), there has been substantial research on the outcomes of environmental and sustainability education. In environmental education, most knowledge-based programs have delivered weak results with respect to changing learners' behaviors. Indicatively, in their book *The Failure of Environmental Education*, Saylan and Blumstein (2011) discuss the generally poor outcomes of environmental education programs in the United States [12]. In a more recent study, Stern, Powell, and Hill (2014) conducted a meta-analysis of literature reporting on the evidence based outcomes of eighty six environmental education programs (or groups of programs)—again, most of the environmental education programs that they analyzed took place in the United States [24]. Even though the programs they reviewed were shown to be effective in improving learners' environmental knowledge, gains in knowledge could not be directly linked with immediate improvements in environmentally responsible behavior [24]. Indeed, while 82% of the analyzed programs report positive outcomes in environmental knowledge, a mere 16% of the programs report positive outcomes in improving learners' environmental behavior (pp. 10–11). On the other hand, those programs that included a strong outdoor component were more effective in changing learners' behaviors as compared to indoor, knowledge based programs (p. 15).

Indeed, a substantial body of empirical evidence suggests that the most effective way to support pro-environmental behavior is outdoor, experiential learning [25–28]. This is consistent with the finding that outdoor experience during childhood is the strongest predictor of adult environmental concern [29,30]. Besides, it has already been established that outdoor experiences during childhood have multiple benefits for individuals' mental and physical health [14,31]. Conversely, children surrounded by low amounts of green space have up to a 55% higher risk of developing a mental disorder in their later lives – even after adjusting for other known risk factors such as socio-economic status, urbanization, and the family history of mental disorders [32]. Besides, active and experiential learning in real-world environments has been shown to directly support learners' pro-environmental behavior in meso-scale experimental designs [33,34].

Hence, the relevant literature supports that experience-based learning has been shown to be more effective in changing learners' behaviors than knowledge-based learning. Experiential learning starts from learners' experience of their communities and local environments. Moreover, experiential learning is a form of learning that aims to apply learners' experience in finding solutions for real-life problems that are affecting the local communities. On these grounds, the exemplary cases from the Global South that were selected to be

presented in this review have a common thread in that they employ experiential rather than knowledge-based learning.

We begin this review by presenting an early case from Colombia, where experiential learning has been effectively employed to meet the place-specific needs of the local communities. This community-based environmental and sustainability education model was documented by Talero and Humaña [35]. In the educational model espoused by this program, the school becomes the center of the community's social and environmental development. Through a participatory approach, the educators call on parents and other members of the community to identify the problems of their locality and its development needs. A conceptualization and implementation phase follows, which sets in motion projects to resolve these problems from an ecological and active community development perspective, including economic aspects; one of the solutions involves the production and processing of pesticide-free fruits by using domestic compost as fertilizer. The Colombian example capitalizes on the cultural resources of its local community by incorporating indigenous and traditional ecological knowledge in its content. These forms of knowledge constitute an ensemble of pre-colonial, place-based learning practices on how to educate for a sustainable future [11,36,37]. Lucie Sauvé (2005) has characterized this holistic approach that incorporates elements of traditional ecological knowledge and place-based learning as a bioregional pedagogical mode of environmental and sustainability education [38].

In a post-hoc evaluation, Mejía-Cáceres and colleagues (2020) characterized the Colombian program as a "very rich example of the national institutionalization of the field . . . an explicit vision of the construction of an 'environmental culture' (of an essentially ethical nature)" [39]. Importantly, this educational program does not restrict its content to purely environmental issues; instead, it involves the local economy, society, and environment in the teaching and learning process. This implies a transformational orientation, where educators are "primarily concerned with generating knowledge of action strategies and providing evidence for how to intervene in real-world problems in order to resolve or mitigate them" [40]. Indeed, knowledge of action strategies was found to be one of the most effective factors in influencing learner's environmental behaviors [41]. In that sense, the Colombian initiative can be characterized as a par excellence Education for Sustainable Development program, where (according to UNESCO's definition) teachers are expected to use "participatory teaching and learning methods that motivate and empower learners to change their behaviour and take action for sustainable development" [11].

The second example presented in this work is an Environmental Education program offered to public school students by a peripheral Environmental Education Centre in Greece. Is it pertinent to consider Greece a part of the Global South? Indeed, a school of thought in post-colonial theory supports the methodological choice of considering Greece as part of the Global South. Indicatively, de Sousa Santos has suggested that southern European countries can be understood by means of a Global South typology. In his work Epistemologies of the South and the future, de Sousa Santos (2016) evokes the internal colonialisms between North Europe and South Europe to explain how Southern Europe eventually became "a periphery, subordinated in economic, political, and cultural terms to Northern Europe and the core that produced the Enlightenment (a condition made more evident with today's financial crisis)" [22].

The Greek educational intervention has a day-long duration and is offered to secondary students by a local Environmental Education Centre that serves public schools of the Peloponnese, Greece [42]. The program includes a visit to a local hydroelectricity dam and its environs, where students are invited to observe the explicit and implicit effects of damming on the natural environment. Through participatory, cooperative and problem-based pedagogies, the students realize that a hydro dam is an intervention with interconnected, and ambivalent consequences. At the closing of the educational program, the students are asked to debate and reach an informed decision on a hypothetical proposal for the construction of a hydro-dam in their area.

The end goal of this and similar programs is to equip students with the knowledge, skills, attitudes, and values that they will need to take part in decision-making that affects their local environments. In this interdisciplinary program, the environmental educator integrates basic elements of geography, ecology, hydrology and soil science, environmental chemistry, economics, engineering, and planning in their teaching. Adapting to the educational level of the students, the instructor exposes the complexity of environmental issues and the role of conflicting interests in any major or minor environmental intervention. Hence, during the program's implementation, the educators cross the barriers between disciplines so often that they eventually cease to notice. Thus, this environmental education program cannot be confined to a conventional subject-based course such as geography or physics, and as such, it is inherently interdisciplinary [43].

While the Greek program has been commended for seeking to include aspects from different disciplines in its content, it does not go into the same depth in involving the local community as the Colombian program does. For that matter, the Greek program is perhaps accurately described as an environmental education program—audibly interdisciplinary, even integrative at times, but without actively involving the local community in problem solving. Besides, the subtle difference between Environmental Education and Education for Sustainable Development is that the latter sets out to explore the broader context of social and political issues vis-à-vis the natural environment [11].

The last case to be presented comes from Mexico, where an Education for Global Citizenship program has been effectively employed to meet the place-specific needs of the local communities. The Education for Democratic Citizenship program was launched in March 2012 in order to address the problem of endemic violence in Nuevo León, Mexico, and has had a positive reception by the local community [44]. The program seeks to engage all members of the school community—students, teachers, administrators, support staff, and parents/guardians—in the educational intervention and convey its core principles in classrooms, school environments, and within the families of the students. Whenever Education for Democratic Citizenship is introduced to a new school community, the basic curriculum is adapted to the specific needs and realities of that community. The Education for Democratic Citizenship program employs a purpose-driven, interdisciplinary pedagogy which encompasses broad domains of knowledge, such as history, literature, music, arts, and pedagogy in order to achieve its end goal.

This specific program from Nuevo León is a multi-stakeholder collaboration that intended to create transcendental solutions that go to the roots of the problem of violence in Mexico rather than simply fighting its symptoms (one of the first proponents of transcendentalist education was, of course, David Thoreau). The objective of the program is to foster students' communicative and organizational skills and their citizenship attitudes such as the value of diversity and tolerance and the rejection of violence as a response to real-life problems [45]. The end goal of the program is to achieve behavioral change, and indeed positive preliminary findings concerning these (non-cognitive) outcomes were reported at the International Conference on Innovative Learning Environments in Banff, Canada [46].

The Mexican program belongs to a series of programs that are funded by the OECD Innovative Learning Environments project, which is being carried out by its Centre for Educational Research and Innovation (CERI) [47]. The umbrella program, Education for Democratic Citizenship, was developed by a non-profit organization that implements innovative educational strategies and intervention models with a systemic perspective to expand the life opportunities of public school students. The umbrella program initiated with the training of 150 teachers from four states in Mexico within urban, rural, and indigenous marginalized contexts [48].

In this section, we presented the central ideas promoted through interdisciplinary education initiatives from Colombia, Greece, and Mexico. Despite the differences in the institutional dynamics and the different degree of participation of private and public stakeholders, there is a common connecting thread, which is that all three programs are

supported by foreign funding. Out of the three programs presented in this review essay, the Mexican program is the one where the national or local government has had the least intervention—the curriculum and program implementation there are run by NGOs. However, the role of NGOs in development and education has received criticism. In particular, it has been supported that as a result of the retreat of centralized government, donors have gained the power to set the agenda through their support to NGOs, which have in turn become a part of the promotion of Western hegemony in the developing world [49].

On the other hand, in the Colombian example NGOs have played an auxiliary rather than a central role. The funding of the Colombian program is based on intergovernmental cooperation instead of NGOs. Indeed, the Colombian program was supported financially by a "debt for environment and social swap" from Canada at the Rio Conference in 1992. This funding channel, managed by the government of Colombia, has brought together NGOs, university and government representatives, as well as a network of regional environmental organizations [50]. However, the Colombian program has also received criticism exactly because of this financial support from countries such as Canada. This external support has been described as "foreign interference", to the extent that the funds were selectively directed to specific academic circles and regions of Colombia [51]. There is a recurring argument according to which the acceptance of external funding in support of educational initiatives cedes political influence to external economic or institutional actors.

The Greek environmental education program has also received foreign funding: the program was developed by a local Environmental Education Centre, which (as part of a national network of fifty-three Environmental Education Centres) receives systematic external funding by the European Union [52]. Despite this source of external funding (or perhaps exactly because of that), it is asserted that Greek Environmental Education Centres enjoy an enhanced level of pedagogical autonomy, where each center is free to diversify its funding sources and develop its curricula and affiliations [53]. A multilateral, horizontal agreement between local governments, the national government, and the European Union secures the infrastructure, staff, and operating costs of Environmental Education Centres in Greece [54].

As earlier discussed, the actuality of educational programs based on foreign funding has given rise to critics who have discovered ensuing tensions between policy, practice, and research. It is not within the scope of the current article to discuss the broader political problematique on whether it is appropriate for educational initiatives in the Global South to accept financial support offered by more affluent countries and/or international organizations. However, given the urgency and the criticality of the problems that environmental, sustainability, and global citizenship education intend to address, it could be supported that foreign funding is warranted, under the condition that the pedagogical autonomy of the local educational initiatives is protected. Ideally, local educational initiatives should be able to mobilize resources from different levels of governance while allowing for an independent curricular planning that serves the educational needs of the local communities.

Hence, without forgetting the caveat inherent in receiving funding derived from powerful economical entities, perhaps it is possible to work for ethical and financial accountability while protecting the political autonomy and the place-based character of the local educational initiatives. A multilateral cooperation between local and national governments, non-profit organizations, small private providers, and external, institutional actors might constitute a balanced scheme—one that would not cede excessive powers to any of the involved parts—in order to protect the long terms interests of the local communities. Besides, it must not escape us that the programs presented in this review essay intend to return their results to the local communities. These are problem-based programs—they center on drawing the connections between different fields of knowledge and intend to inform and mobilize the local communities into finding solutions to their real-world problems. Even if the observed programs have different degrees of success in

involving the local communities, the experiential rendition of Environmental, Sustainability, and Global Citizenship Education always starts from the places where we live, work, and study.

## 3. Challenges for Interdisciplinary Learning

As a result of academic and political advocacy, a number of countries have introduced Environmental Education, Education for Sustainable Development, and Education for Peace and Global Citizenship in their curricular planning. As discussed in the previous section, these interdisciplinary educational approaches have enjoyed strong political support by international organizations. However, as implied earlier, several real-world difficulties remain to be addressed before these initiatives develop into comprehensive educational proposals. This section elaborates on the pragmatic challenges that these integrative schemes have met upon their application in actual educational settings.

We have indeed seen that all three programs featured in this review essay involved interdisciplinary and experiential teaching and learning methods with participants' senses and experiences from their local communities. However, it is well documented that interdisciplinary teaching approaches have encountered difficulties upon their mainstreaming into school curricula. The literature suggests that on the school-unit level, interdisciplinary approaches are often perceived as incompatible with the existing curricular practice of dividing content knowledge to presumably independent subject-matters. Both in terms of content and time-schedule, interdisciplinary learning is still striving to find its niche. The most common difficulty associated with the introduction of interdisciplinary learning approaches is the inflexible school schedule itself. Indicatively, in a study from one of the featured countries, the majority of in-service teachers cite the the strict school timetable as the most important impediment for the implementation of environmental education programs [55].

This is not merely a technical issue—teachers report that they do not feel comfortable to work on subjects matters that are distanced from their own specialties [56]. In a nutshell, interdisciplinary pedagogies are not always seen as compatible with the contemporary subject-based organization of knowledge. Since interdisciplinarity is essential to Environmental, Sustainability, and Global Citizenship Education, further conflict between these and the current educational ethos is to be expected. A clear understanding of the institutional dynamics that are at the root of the prolonged resistance to the implementation of integrative pedagogies will facilitate future planning by offering a realistic outlook on the difficult challenge of educational transformation.

With no intention to underestimate the pertinence and agency of strong political will in promoting educational change, the following section aims to explore the presence of deeper, epistemological issues that may account for the slow progress of these teaching and learning approaches. We hereby present the argument that the main cause of institutional reaction to the introduction and dissemination of interdisciplinary educational approaches is not political, but epistemological. Hence, according to the main argument presented in this essay, the most persistent challenges—and also the most difficult to address—are of epistemological nature.

## 4. Epistemological Barriers to Interdisciplinary Practice

The interdisciplinary nature of Environmental, Sustainability, and Global Citizenship Education challenges the contemporary ethos of dividing knowledge into separate subject matters. However, the current practice by which the scholarly community creates, maintains, and reproduces knowledge is based on an educational system where the curriculum is organized in subjects. For example, geography is one of the courses taught at school; poetry and science are taught as distinct, unrelated courses. In that respect, an Environmental Education initiative aiming to develop "geography students' understandings of sustainable development using poetry" [57] might be raising some eyebrows even today. These initiatives, however, respond to an educational need: today, it is increasingly

understood that high complexity problems, such as the interactions between ecosystem components and their interdependence with the functions of social and political networks, require extensive dialogue between different fields of knowledge [58,59]. If that is the case, then we need a kind of education that would encourage interdisciplinary cooperation and synthesis in its teaching and learning practices.

However, this is not going to be an easy task, the reason being that the classification of knowledge into separate "siloes" is a deeply embedded practice for contemporary educational systems. Discussing the difficulties encountered by environmental education programs in Brazil, Paula Brügger notes that today, knowledge is compartmentalized to the extent that adjacent disciplines are unable to establish a meaningful dialogue between them [15]. As far back as 1929, mathematician and educational reformist Alfred Whitehead has deplored the "fatal disconnection of subjects which kills the vitality of our modern curriculum" [60]. The current fragmentation of academic disciplines into disconnected subjects is attributed to the impact of scientific reductionism, which has historically created and maintained insulated epistemic traditions [61]. Scientific reductionism was introduced after Descartes' (1662) suggestion that complex systems can be studied by their reduction into isolated parts [62]. The legacy of scientific reductionism in education is the fragmentation of knowledge into mutually exclusive fields of expertise.

Education was not always been delivered by dissociated, mutually exclusive subject matters. The ancient and medieval educational paradigm was based on the seven liberal arts—and was holistic in nature and practice [63]. All seven arts were virtually "ways of doing" or skills which were considered as the essentials for the development of a free person [64]. The core arts of grammar, logic, and dialectic were used as general framework of reasoning that found application in the practical arts of arithmetic, geometry, astronomy and music. The latter were practical in the sense that they were associated with tasks such as measuring the land (geometry) or organizing time (astronomy). Thus, the knowledge system of antiquity placed what we today call "the humanities" at its core and considered practical arts as its applications [65]. After the Enlightenment, the taxonomy of knowledge changed from skill-based to content-based and knowledge fields were re-organized according to subject matters. Henceforth, knowledge was organized in the contemporary form of distinct subject matters and the connections between knowledge disciplines gradually waned. The reorganization of knowledge according to subject-oriented produced a fragmented universe of academic disciplines, which tends to create and maintain insulated research traditions. However, it is increasingly been understood that this practice does not meet the contemporary educational needs of today, which require a greater emphasis on learners' skills of interdisciplinary and synthetic thought.

This essay proceeds by reviewing recent research practices in search of interdisciplinary efforts that challenge the existing ethos of dividing knowledge into separate, presumably independent, subject matters. On these grounds, it maps academic research to reveal loci where interdisciplinary cooperation and synthesis are currently taking place, most notably in the fields of ecology and environmental studies. In so doing, it raises the question: Is there a need for further interdisciplinary synthesis? If yes, does interdisciplinary synthesis meet epistemological resistance? Can interdisciplinary dialogue act as a catalyst for the transition to a new, integrated hermeneutic paradigm?

## 5. Something Is Changing in Knowledge Production

Starting in the 1970s, a number of epistemic initiatives, most notably from the field of ecology, have ventured to combine elements from different fields of knowledge in order to address interdisciplinary problems [66]. Indicatively, the shared usage of terms and concepts pertaining to sustainability, resilience, carrying capacity, etc., between academic fields is a good example of the role of ecology in enabling interdisciplinary dialogue [67]. The position of ecology and environmental science as conceptual bridges the visual of Figure 1 is consistent with their role as pioneers of interdisciplinarity. The very subject of ecology, the study of complex interactions between biotic and abiotic environmental

components, allows it to act as a connection between different academic disciplines. In that sense, ecology has paved the way for interdisciplinary efforts in education, starting from Environmental Education. However, despite the success of interdisciplinary efforts in creating a shared vocabulary and facilitating the exchange and adaptation of individual methods between fields, these have not hitherto proceeded into genuine interdisciplinary or transdisciplinary synthesis [68]. The contemporary taxonomy of knowledge still lacks the integrated framework that would allow it to address the problems of environmental and social sustainability in their full depth and complexity. The complex problems of today, with all their social, political, economic, and environmental ramifications, necessitates a de novo holistic approach. Besides, according to David Orr (1992) "[t]he symptoms of environmental deterioration are in the domain of natural sciences, but the causes lie in the realm of the social sciences and humanities" [69].

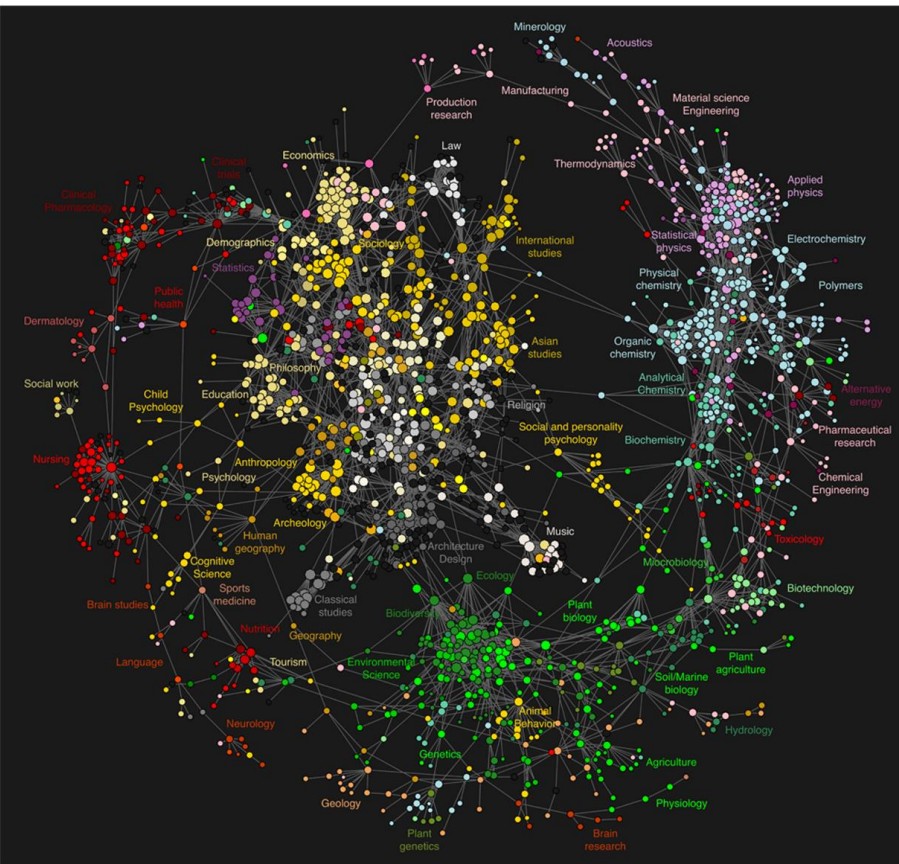

**Figure 1.** Ecology and Environmental Science as catalysts of interdisciplinary synthesis. In order to construct an affinity map of academic disciplines, Bollen et al. (2009) have amassed and integrated 1 billion ($1 \times 10^9$) user interactions logged by web portals operated by renowned scholarly networks such as Elsevier (Scopus), JSTOR, and Thomson Scientific (Web of Science) [70]. In the generated visualization of contemporary research, the division between science and the humanities becomes apparent. The humanities and related fields form a lobe that occupies the central part of the graph, while science and relevant disciplines form an outer lobe in the right part of the graph. This is only one of the equivalent representations of contemporary research, which confirm what Charles Snow described as the "two cultures" of science and the humanities back in 1959 [71]. Although a certain degree of osmosis occurs today between the "two cultures", the two lobes on the graph are well defined and encompass minor sub-constellations of disciplines. Most of the interconnections between the two lobes pass through the conceptual bridges of ecology and environmental science. Figure released by Bollen et al. (2009): https://journals.plos.org/plosone/article?id=10.1371/journal.pone.0004803 (accessed on 14 August 2021).

### 6. The Role of Education in Achieving Transformative Change

The type of interdisciplinary cooperation promoted by Environmental, Sustainability, and Global Citizenship Education poses a transformational requisite for our teaching and learning traditions, and as such, it cannot be viewed separately from the dominant postulations, practices, and priorities of educational systems [72]. Indeed, delineating the move towards the UN Sustainable Development Goals UNESCO claims to be "supporting countries in making this transformative change" [73]. Moreover, UNESCO describes its bid for transformative change in education, mandated by the UN Sustainable Development Goals, as a "paradigm shift" [11]. These are ambitious goals set out by UNESCO to be carried out by the institutional providers of education.

Prompted by this choice of words, this article explores the epistemic origins of the "paradigm shift" terminology and hence sets out to examine whether the aforementioned interdisciplinary pedagogies meet the Kuhnian conditions to be considered as new candidates for paradigm in the field of education. Since UNESCO opts to employ a paradigm shift terminology, we ponder: Could it be that the interdisciplinary approaches espoused by international organizations portend a paradigm shift in the Kuhnian sense? In other words, could environmental, sustainability, and global citizenship education be the precursors of a paradigm shift that will reconnect the knowledge scattered over diverse fields of scholarship under a unified conceptual system? Or are these educational schemes lacking in substance and will most likely degenerate and be forgotten as ephemeral trends that only managed to involve a limited audience? Before moving on to address this question, it is necessary to examine the applicability of Kuhnian theory in education—this task is performed in the following section.

In his theory on the structure of scientific revolutions, Thomas Kuhn (1962) uses the term "paradigm shift" to describe processes where a new paradigm of epistemic thought succeeds an old paradigm [74]. Kuhnian theory has produced itself a paradigm shift in the field of epistemology by explaining how science—and, more broadly, shared hermeneutic schemes—alternate between long periods of stability and short periods of abrupt, revolutionary change; this observation challenged the positivist view of science as being a steady progress towards an objective truth. Furthermore, since Kuhnian theory pertains to cosmologies or ways of knowing, it has been used before in a scope of application much broader than the evolution of ideas within western science. See, for example, Stanley Ivie's work (2017), where he applies Kuhnian theory to identify of six successive historical periods of epistemic cosmologies that span over twenty-five centuries of written history [75].

Following a somewhat narrower scope of application, the present study uses Kuhnian theory to distinguish the systems of thought (or conceptual systems) that have been historically endorsed by different cosmologies (Figure 2). The field of education is directly linked to the systems of thought which inform the different methods of teaching and learning. Education teaches ways of knowing or ways of seeing the world—eventually, education teaches learners how to narrow down on their "incommensurable ways of seeing the world" and select the worldview which is favored by each historical lieu [74]. Hence, education is one of the fields where Kuhnian theory can be (and has been) applied.

Indeed, Kuhnian philosophy explicitly acknowledges the role of education in shaping paradigms or worldviews. Kuhn identifies the route that connects stimulus to sensation as central in providing explanations of the world and highlights that this route is "partly conditioned by education" [74]. In reaction to identical stimuli, individuals raised in different societies behave on some occasions as though they saw different things. He expounds that two groups, the members of which have systematically different sensations on receipt of the same stimuli, do, in some sense, live in different worlds. Thus, differences in perception are central in identifying the difference between paradigms, and paradigm shifts cannot occur without respective changes in the route that connects stimulus to sensation. According to Kuhnian theory, one of the methods to identify whether a paradigmatic shift is under way is to look out for changes that affect the route connecting stimulus to sensation as conditioned by education.

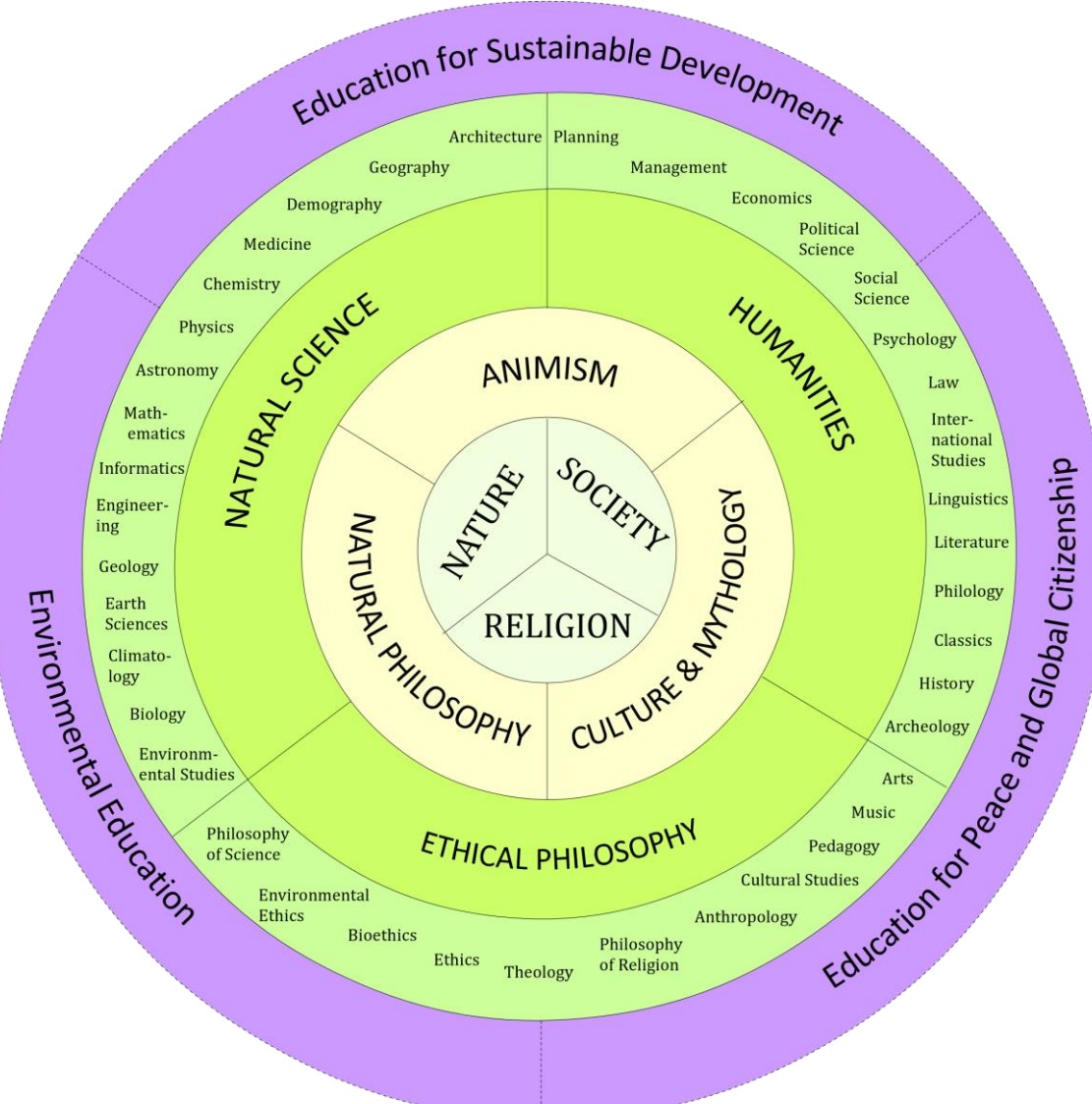

**Figure 2.** An explanatory scheme on the epistemic evolution of conceptual systems. This is a heuristic visualization of the continuing evolution of conceptual systems as typified in $C_{3v}$ point group symmetry. The inner circle, composed of three parts, constitutes an ontological definition of the cosmos as comprised of the social, natural, and supernatural elements. The successive homocentric circles represent the epistemic evolution of conceptual systems from the ancient-medieval, to the modern, and eventually to the post-modern taxonomies of knowledge.

## 7. Conditions for Paradigm Shift in Education

After arguing for the applicability of Kuhnian theory in education, we venture to analyze what the theory generates in the selected scope of application. This section employs a macroscopic application of the Kuhnian theory in order to explore whether Environmental Education (EE), Education for Sustainable Development (ESD), and Global Citizenship Education (GCE) fulfil the necessary (if not sufficient) conditions to be considered as candidates for a paradigm shift in education. As is the case with any theoretical framework, the analytical power of Kuhnian theory produces different results depending on its scale of application. According to Kuhn's (1962) theory on the structure of scientific revolutions, epistemic progress is characterized by the succession of different paradigms (i.e., world-views) [74]. In Kuhn's words, "A paradigm is an accepted model or pattern" of ideas, around which a professional community organizes its thinking [74]. Paradigm shifts happen when a professional community encounters an outstanding and generally recognized problem

that can be addressed in no other way. Kuhn has set two all-important conditions that need to be met before a scholarly community can embrace a new paradigm.

First, "the new candidate must seem to resolve some outstanding and generally recognized problem that can be met in no other way" [74]. From an educational perspective, the outstanding problem is that there is a growing demand for interdisciplinary skills which cannot be met by the established educational ethos of subject-based curricula and thematically separated fields.

Hence, as specified in education, the Kuhnean outstanding problem is the apparent inadequacy of the educational system to provide its recipients with the conceptual tools that are necessary for addressing the environmental crisis. If we accept the premise that (a) the environmental crisis is anthropogenic, (b) education is an important factor in shaping our behaviors towards the natural environment, and (c) the environmental crisis is solvable, it entails that education is from where we should be starting in order to address the problem.

In that respect, the Kuhnean outstanding problem manifests in the field of education through the conflict between the current practice of subject-based curricula and the inter-disciplinary approaches that are deemed as necessary to address the complex social, ethical, and economic implications of environmental degradation. The case has been made that the disconnection between academic disciplines is an impediment for the complex way of thinking that solutions to environmental problems require. Hence, from an educational perspective, the outstanding Kuhnian problem is that contemporary education systems do not equip their students with the interdisciplinary skills that are becoming increas-ingly necessary, not only for their professional standing, but also for their everyday ability to navigate through systems of increased complexity. The inadequacy of contemporary systems of thought in addressing complex contemporary problems, with environmental degradation as the most prominent of them, has been discussed throughout this article. The interdisciplinary educational approaches of ESD, EE, and GCE do promise to resolve an outstanding and generally recognized problem in education and thus meet the first theoretical condition of a Kuhnian candidate for paradigm.

As a second condition, Kuhn [74] has projected that the new candidate for paradigm must promise to preserve a "relatively large part of the concrete problem solving ability that has accrued to science through its predecessors". Thus, in order to be considered as candidates for paradigm, ESD, EE, and GCE should equip students with the conceptual tools necessary to preserve a large part of the problem solving ability of natural science, social science and humanities combined (Figure 2). This is not an easy task, given that con-temporary education systems, based on the separation between science and the humanities, have accrued extensive methodological traditions that have proven their solving ability in diverse types of problems. However, the historical preference of these realms of knowledge for different methodological tools created diverging research ethoi and the communication between these siloed knowledge domains has become problematic. While the education of students on specialized subject matters has accrued extensive methodological knowledge of problem-solving within these strictly defined fields of expertise, it is not compatible with the interdisciplinary thinking that complex contemporary problems require. Eventually, the question of conjunctive problem-solving ability comes down to the challenge of combining independent methodological systems. Hence, the synergy of different methodological traditions is a prerequisite for the preservation of the problem-solving ability of the present educational paradigm.

Is it possible to promote educational systems that would fuse different methodologies without losing the greatest part of its constituents' problem-solving ability? Based on the available evidence, it is hard to answer the question of whether interdisciplinary efforts can preserve a relatively large part of their predecessors' problem-solving ability. Interdis-ciplinary dialogue and synthesis have just begun (in terms of historical time scales) and the methodological tools of these interdisciplinary efforts are not yet established. However, recent developments allow for the integration of methods from different methodological schools that were previously viewed as incompatible [76]. Mixed methods research designs,

which seek to reconcile different methodological traditions, constantly gain ground in ESD, EE, and GCE pedagogical practice [24]. Hence, integrated methodologies are increasingly comfortable in applying a customized amalgam of qualitative and quantitative methods that are focused on the pragmatic needs of each research question. In educational practice, the relevant body of research shows that experiential and place-based methods provide excellent opportunities for introducing learners in the interdisciplinary ways of thinking that are required in order to resolve the actual problems of the communities where they live, work, and study [77]. Recent research explores the question of how connectedness to nature can be taught and mainstreamed in school-based educational practices [78,79].

In the field of ecology, interdisciplinary dialogue started as early as the early 1970s [66]. As a result, the borders between the academic silos have become more porous [80–82]. Thus, by establishing a meaningful dialogue between different methodological traditions, the field of ecology seems to have acted as a precursor of interdisciplinary synthesis. The very subject of ecology, which is the study of complex interactions between biotic and abiotic environmental components, allows it to act as a conceptual bridge between different academic fields (Figure 1). Hence, it not only education that has become more interdisciplinary in its goal setting, but also science itself that is moving toward interdisciplinarity.

Concluding this section, the effort to fuse methods from different research traditions is only in its start, and extensive methodological work is still needed. Eventually, the effectiveness of the methodological toolkits promoted by ESD, environmental education, and GCE will determine whether these educational initiatives are here to stay. If the new educational initiatives preserve a significant part of the problem-solving ability of the current paradigm, they may become constituents of the next paradigm. Alternatively, these initiatives may constitute a transient form of Kuhnian anomalous science that will lead to a new paradigm of yet unknown nature. At present, we do not have enough evidence to conclude whether the methods espoused by ESD, EE, and GCE preserve a significant part of the problem-solving ability of the existing educational paradigm. It should not escape us that ESD, EE, and GCE are tentative educational fields and there is more work ahead in order to crystallize their practices, methods, and objectives. In the past, similar trans-disciplinary efforts have failed to achieve a sustained and lasting impact.

## 8. Conclusive Remarks

This review article discussed the theoretical underpinnings of environmental, sustainability, and global citizenship education drawing on three actual cases from Greece, Colombia, and Mexico, respectively. All three examples represented approaches to education that are interdisciplinary in their nature, and interdisciplinarity is a practical challenge in its own right. Education for Sustainable Development brings together disciplines from natural science and the humanities under the common cause of sustainability. Environmental Education has a deeper aim in bringing science in a dialectic relationship with ethics and, beyond that, create a feeling of respect and reverence towards the natural environment. Education for Peace and Global Citizenship bears the promise of reconciling ethics with social practices by addressing issues such as social and environmental justice. Being interdisciplinary in their nature, these broad educational approaches are more accurately defined by their purpose rather than their content (Figure 2).

A macroscopic application of Kuhnian theory introduced a notion of historical succession of conceptual systems from skill-based (ancient/medieval) to content-based (modern) to purpose/problem-based (postmodern). A purpose-based taxonomy of knowledge is notably distinct from the current division of knowledge into subject matters and its older classification according to practical skills. In that sense, Environmental, Sustainability, and Global Citizenship Education were assigned a task that is greater than their own potential – which has to do with how we practice science as a whole. This might explain the difficulties encountered: interdisciplinarity is harder challenge than originally assumed, because it challenges the core contemporary assumptions by which we maintain and reproduce our knowledge systems—a problem of epistemological nature.

As these novel approaches strive to build their epistemological identity, the urgency of the environmental problem (with its socio-economic underpinnings) might curb systemic resistance to interdisciplinary dialogue and integration. However, it should not escape us that these are emergent fields, and there is more work ahead in order for them to secure their epistemological niches. The future success of the three pedagogical initiatives will depend on whether they will be able to preserve an adequate part of the problem-solving ability accrued by their predecessors. At present, we do not have enough empirical evidence to suggest whether these integrative approaches can undertake the task of a deeper epistemic transformation. Further research and cross-disciplinary dialogue are needed before we can reach a certain level of agreement on the extent to which the integration of knowledge systems is feasible, desirable, and necessary.

**Funding:** This research was supported by the Stavros S. Niarchos Foundation, the Fonds de recherche du Québec—Société et culture (FSQSC), grant number 201395, and the State Scholarships Foundation, grand number 2012-ΠΕ2-1844.

**Acknowledgments:** I would like to acknowledge the contribution of Sylvie Wald who, when we were both studying at McGill, shared her outlook on the ethical systems that historically coexisted in East Asia, namely Confucianism, Taoism, and Buddhism. Her understanding of these constructs as regulators of peoples' relationship with the social, natural and metaphysical realms helped me in developing the explanatory scheme on the evolution of conceptual systems that is being presented in Figure 2 of this essay. Also, in last-name alphabetical order, many thanks to: Anila Asghar, Omar Aziz, Merve Erdilmen, Heidi Hoernig, Pascal Kropf, Sabeena Shaikh, Aristofanis Soulikias, and Ralf St.Clair for their valuable feedback and comments. Special thanks to IT experts Angeliki Athanasiou and George Koumparoulis for their quick action in recovering this manuscript when its code went corrupt.

**Conflicts of Interest:** The author declares no conflict of interest.

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
