# Peer review of "Education for Sustainability, Peace, and Global Citizenship: An Integrative Approach"

_education, doi:10.3390/educsci11080430_

Round 1
Reviewer 1 Report
A very interesting article combining theory and cases from three countries (from the Global South) in order to discuss the reasons behind the slow incorporation of interdisciplinary approaches (like Environmental Education, Education for Sustainable Development, and Education for Global Citizenship) into actual education settings. Based on the presentes cases, this discussion offers pragmatic solutions in order to promote the integration of those approaches into the existing educational ethoi.
The discussion is very inspiring, interesting and formative for Education Sciences readers, regarding the implementation of Environmental Education, Education for Sustainable Development, and Education for Global Citizenship, into actual education settings.
Author Response
Thank you for bothering to read my manuscript and for your encouraging comments!
Reviewer 2 Report
It's an essay written on an important topic. It indeed a welcoming essay to encourage interdisciplinary approach to the environmental, sustainability and global citizenship education. The visualizations of data points were immensely helpful.
It's a professionally written piece of work. however, I have a few questions that the author might consider.
Please add more literature as I think there will be many studies that explores the interdisciplinary nature of the topics that you explored in this essay, especially some recent (after 2019 until now) academic literature in this field will be helpful. Although the issue is very current and you have done a great job of bringing those studies to light, including some of this would strengthen the article and ground it in the literature.
I think in line 485 to 488 you make an important claim, but evidence needs to add there to support that claim. Or is it something you are predicting? So, it's unclear.
Although it's an essay, however it would be nice to see some discussion in this essay on clear recommendations for next step, I know you say that there is more evidence require, but is there any practical implication for anyone or society?
Author Response
Please open the attached cover letter.
